# First Modern Data on the Lophophore Nervous System in Adult *Novocrania anomala* and a Current Assessment of Brachiopod Phylogeny

**DOI:** 10.3390/biology11030406

**Published:** 2022-03-06

**Authors:** Elena Temereva

**Affiliations:** 1Department of Invertebrate Zoology, Biological Faculty, Moscow State University, 119991 Moscow, Russia; temereva@mail.ru; Tel.: +7-(909)-9764434; 2Faculty of Biology and Biotechnology, National Research University Higher School of Economics, 101000 Moscow, Russia

**Keywords:** Brachiopoda, nervous system, lophophorates, lophophore, evolution, phylogeny

## Abstract

**Simple Summary:**

The nervous system of *Novocrania anomala* adults is described for the first time. A table containing data on the lophophore innervation in species from three brachiopod subphyla is presented. A comparative analysis suggests a close relationship between the Craniiformea and the Rhynchonelliformea, and thereby supports the “Calciata” hypothesis of brachiopod phylogeny.

**Abstract:**

Although the lophophore is regarded as the main synapomorphy of all lophophorates, the evolution of the lophophore in certain groups of lophophorates remains unclear. To date, the innervation of the lophophore has been studied with modern methods only for three brachiopod species belonging to two subphyla: Linguliformea and Rhynchonelliformea. In the third subphylum, the Craniiformea, there are data for juveniles but not for adults. In the current research, the innervation of the lophophore in *Novocrania anomala* adults was studied by immunocytochemistry and confocal laser scanning microscopy. In the spiral lophophore of adults of the craniiform *N. anomala*, each arm is innervated by six brachial nerves: main, additional main, accessory, second accessory, additional lower, and lower brachial nerves. Compared with other brachiopod species, this complex innervation of the lophophore correlates with the presence of many lophophoral muscles. The general anatomy of the lophophore nervous system and the peculiarities of the organization of the subenteric ganglion of the craniiform *N. anomala* have a lot in common with those of rhynchonelliforms but not with those of linguliforms. These findings are consistent with the “Calciata” hypothesis of the brachiopod phylogeny and are inconsistent with the inference that the Craniiformea and Linguliformea are closely related.

## 1. Introduction

Brachiopods are benthic marine animals that form a phylum consisting of 400 recent species and more than 10,000 extinct species. All brachiopods have dorsal and ventral shell plates, which are produced by the body wall (mantle) and cover the soft body. According to the most common phylogeny, all recent brachiopods can be subdivided into three subphyla: Linguliformea, Craniiformea, and Rhynchonelliformea [1,2,3]. The first two subphyla are traditionally gathered into “inarticulate” brachiopods, because their shell valves lack articulate structures. According to some data, the clade of “inarticulate” brachiopods includes phoronids, which are the closest relatives of brachiopods; moreover, phoronids are regarded as “brachiopods without shell” [4]. This close relationship is supported by many molecular data, and the clade, which includes brachiopods and phoronids, is called Brachiozoa [5,6,7]. The shell valves of brachiopods in the subphylum Rhynchonelliformea have articulate structures and are traditionally regarded as “articulate” brachiopods. The division of Brachiozoa into “articulate” and “inarticulate” groups has been supported by some data on molecular phylogeny [4]. At the same time, the chemical structure of the shell, anatomy, and development differ greatly among “inarticulate” brachiopods. Thus, linguliforms have the organic chitinous phosphatic shell, whereas craniiformes have calcic shells [8,9]. In linguliforms, a pelagic juvenile presents in the life cycle, whereas craniiformes lack such a juvenile and have bilobed lecitotrophic larvae [10,11]. Such larvae are supposed to form most of the fossil brachiopods [12]. The contradiction between different opinions makes brachiopod phylogeny unclear [13]. The recent phylogeny of brachiopods may be considered in the frame of four main hypotheses: “Inarticulata hypothesis” versus “Calciata hypothesis” and brachiopod monophyly versus brachiopod paraphyly in respect to phoronids [13]. According to “Calciata hypothesis”, craniiforms and rhynchonelliforms with calcic shell are close relatives.

Brachiopod evolution is closely related to the transformation of the lophophore [14]. The lophophore is a tentacular organ that is present in all lophophorates: phoronids, brachiopods, and bryozoans. The monophyly of lophophorates has been recently rebuilt by morphological [15,16,17,18,19,20,21] and molecular [22,23,24] data. The morphological data, which mostly concern the organization of the lophophore nervous system, support the homology of the lophophore in all three phyla of lophophorates. At the same time, differences in the evolution of the lophophore in each phylum of lophophorates are still discussed [25,26,27,28].

In adult brachiopods, the organization of the lophophore nervous system has been studied by immunocytochemistry and confocal laser scanning microscopy in three species from two subphyla: Linguliformea (*Lingula anatina*) [15] and Rhynchonelliformea (*Hemithiris psittacea* and *Coptothyris grayi*) [27,28]. These studies revealed two trends in the evolution of the lophophore in brachiopods. First, a reduction in the main lophophore nerve elements has been accompanied by an increase in the complexity of lophophore morphology [21]. Second, the appearance of a double row of tentacles has been accompanied by an intensification of the second accessory brachial nerve, i.e., by an increase in the complexity of the lophophore nervous system [27]. The first trend is well expressed in rhynchonelliforms, whereas the second trend is typical for most brachiopods.

In adult craniiformes, the organization of the lophophore nervous system has not been studied by immunocytochemistry or confocal laser scanning microscopy. Such data concerning the organization and ontogenic transformation of the lophophore nervous system in craniiformes are limited to juveniles of *Novocrania anomala* [19]. Comparison of previous results [29] and recent data [19] revealed the difference in organization of the nervous system in adults and in juveniles of *N. anomala*. In particular, juveniles have the prominent supraenteric ganglion, which is absent in adult craniiforms. Thus, in the current research, I used immunocytochemistry and confocal laser scanning microscopy to study the lophophore nervous system in *N. anomala* adults. The study of the lophophore nervous system in the adults of a craniiform species will increase our understanding of brachiopod phylogeny and lophophore evolution. It will also fill the gap in our knowledge of the general anatomy of the lophophore nervous system in brachiopods from all three subphyla. Because the organization of the nervous system is traditionally used for establishment of the relationships between different groups, new data will allow a comparative analysis to be made, the results of which may be next applied for the test of different hypotheses about brachiopod phylogeny.

## 2. Materials and Methods

### 2.1. Animals

In the North Sea (Storingavika Bay) in May 2019, a drag net was used to collect stones and adhering adults of *N. anomala*. The animals were carefully separated from the stones, except the ventral valves that remained attached to the stones. Specimens were dissected to obtain lophophores with mouth, tentacles, and brachial fold. Live animals and separated lophophores were photographed with a Leica M165C (Leica, Wetzlar, Germany) stereomicroscope equipped with a Leica DFC420 digital camera. Parts of the lophophores were fixed for semi-thin sectioning, scanning electron microscopy (SEM), and immunocytochemistry combined with confocal laser scanning microscopy (CLSM).

### 2.2. Fixation and Microscopy

For semi-thin sectioning and SEM, lophophores were fixed in 2.5% glutaraldehyde in 0.2 M cacodylate buffer for 8 h at 4 °C. The specimens were then washed in 0.2 M cacodylate buffer for 24 h and were postfixed in 1% osmium tetroxide in the same buffer for 3 h at room temperature. After postfixation, specimens were washed three times for 30 min each in distilled water and were dehydrated in a series of ethanol solutions: 10–30–50–70–96% (10 min for each step). 

For semi-thin sectioning, after dehydratation, specimens were immersed in pure isopropanol for 20 min, then in a mixture of isopropanon and resin—Embed-812 (Electron Microscopy Science, cat. #14120): 3:1 (30 min), 1:1 (24 h), 1:3 (24 h). Then, specimens were infiltrated in pure resin for 24 h at 4 °C. After infiltration, specimens were embedded in resin for 48 h at 60 °C. Semi-thin sections (500 nm in thickness) were prepared with a Leica UC-7 ultramicrotome (Leica Microsystems GmbH, Wetzlar, Germany). Semi-thin sections were stained with 1% methylene blue: 1% C_16_H_18_CIN_3_S × 2-3H_2_O + 1% NaHCO_3_ + 50% saccharose (with distilled water). Semi-thin sections were stained for 3 min at 65 °C. Sections were then observed with a Zeiss Axioplan2 microscope and photographed with an AxioCam HRm camera (Carl Zeiss, Oberkochen, Germany). 

For SEM, lophophores were dehydrated in acetone, critical point dried, mounted on stubs, and then sputter coated with platinum–palladium. Specimens were examined with a Jeol JSM scanning electron microscope (JEOL Ltd., Tokyo, Japan).

For immunocytochemistry, parts of *N. anomala* lophophores were fixed in 4% paraformaldehyde in 0.2 M phosphate-buffered saline (PBS) (pH 7.4) (ThermoFischer, Pittsburgh, PA, USA) for 8 h at 4 °C. After fixation, specimens were washed in PBS with Triton X-100 (10%) (ThermoFischer) (PBT): 8 times for 20 min each. Nonspecific binding sites were blocked with 10% normal goat serum (Jackson ImmunoResearch, Newmarket, Suffolk, UK) in PBT for 24 h at 4 °C. The specimens were incubated overnight in primary antibody (mouse anti-acetylated-a-tubulin; 1:700; cat. #32-2500; ThermoFischer) in PBT at 4 °C, washed in PBT three times for 5 h each, and exposed to the secondary antibody (goat anti-mouse conjugated with Alexa-635; 1:1000; cat. #A-31574; ThermoFischer) in PBT for 2 h at 4 °C. Then, the specimens were immersed in Alexa Fluor-488 conjugated phalloidin (1:50) (ThermoFischer, cat. num. 12379) in PBT at room temperature in the dark for 2 h, washed in PBS (four times for 20 min each), and embedded in Murray Clear (50:50 mixture of benzyl benzoate and benzyl alcohol). Specimens were examined with a Nikon Eclipse Ti confocal microscope (Nikon, Thermo Fisher Scientific, Waltham, MA, USA).

Z-projections were generated using Image J version 1.43 software. Volume renderings were prepared using Amira version 5.2.2 software (ThermoFischer, Waltham MA, USA). Schemes, photographs, and Z-projections were processed in Adobe Photoshop CS3 (Adobe World Headquarters, San Jose, CA, USA).

### 2.3. Terminology

The terminology used in this report is in line with the traditional terminology that was suggested in the first papers devoted to the brachiopod nervous system [29,30]. The traditional terminology suggested by the first papers devoted to the brachiopod nervous system [29,30] is often inconsistent with the recent knowledge concerning the organization of the nervous system [31]. To avoid misunderstandings in the descriptions of the brachiopod nervous system, however, the current paper has used the traditional terminology. Thus, I use the term “ganglion” for description of the subenteric nerve center, which is organized as stratified neuroepithelium, but not as typical ganglion [32].

## 3. Results

### 3.1. Morphology of the Lophophore and Tentacles

The shell of *N. anomala* consists of two valves: the dorsal (=brachial) and the ventral (=pedicle) valves. The ventral valve is cemented to the hard substratum and usually remains attached to the substratum when animals are collected. This separation enables observation of the animal’s ventral side (Figure 1A). Under the dorsal valve, the anterior area is occupied by the mantle cavity, and the posterior area contains the body with prominent muscles (Figure 1A). The lophophore and tentacles completely occupy the mantle cavity, and the tentacles even extend beyond the edge of the shell (Figure 1B). The lophophore is a spirolophe: it has two brachial arms, each of which is spirally coiled (Figure 1C). These spirals extend to the dorsal valve and cannot be observed from the ventral side of the body (Figure 1A,B). The tip of each spiral is the zone where new tentacles form (Figure 1C). The mouth is located between the two brachial arms (Figure 1A and Figure 2A). The mouth extends to the esophagus, which forms a prominent knob above the mouth (Figure 1C and Figure 2A). The mouth is covered by an epithelial fold, i.e., the brachial fold (Figure 2A). One row of oral tentacles is located under the mouth (Figure 2A). Each brachial arm has a large base that contains coelomic cavities (Figure 2C) and bears a double row of tentacles and the brachial fold (Figure 2B, C). The double row of tentacles is formed by inner and outer tentacles. The inner tentacles are located closer to the brachial fold than the outer tentacles (Figure 2B). The food grove is bordered by the brachial fold and the inner tentacles. Inner and outer tentacles also differ in morphology and shape in cross-section: outer tentacles bear a frontal groove, whereas inner tentacles bear a frontal ridge (see below) (Figure 2D and Figure 3A,B). Tentacles of both types have different zones that extend along each tentacle and are characterized by a specific histology of the epithelium and a specific location of muscles and nerves. There are at least four zones: the frontal zone faces the brachial fold, the abfrontal zone is opposite to the frontal zone, and the two lateral zones face the adjacent tentacles. The frontal zone forms a ridge in the inner tentacles (Figure 2D and Figure 3A) and forms a shallow groove in the outer tentacles (Figure 2D and Figure 3B).

### 3.2. Main Nerve Elements of the Lophophore

*N. anomala* adults lack the supraenteric ganglion; only thin dorsal neurite bundles extend above the esophagus (Figure 4A and Figure 5A). These thin neurite bundles connect two branches of the main brachial nerve, each of which extends into each brachial arm (Figure 4A,C). Near the esophagus, each branch of the main brachial nerve bifurcates and gives rise to the circumenteric connective (Figure 4B) that skirts the lophophore base and passes to the subenteric ganglion (Figure 5A). According to CLSM data, the subenteric ganglion looks like a thick nerve, which extends under the mouth and is the largest nerve center in *N. anomala* adults (Figure 4C).

### 3.3. Innervation of the Brachial (Lophophore) Arms

Each brachial arm contains a large and small coelomic canal (Figure 2C and Figure 6A). The small canal is occupied by the circular lophophoral muscle and gives rise to the coelomic canals in each tentacle (Figure 6A). The following six nerves extend along each brachial arm: the main, additional main, additional, second additional, additional lower, and lower brachial nerve tracts (Figure 4C, Figure 5B and Figure 6A). All of these nerves exhibit acetylated alpha-tubulin immunoreactivity (Figure 7; Appendix A). 

**The main brachial nerve** extends along the base of the brachial fold (Figure 6B) and consists of two parts, i.e., the upper and lower parts (Figure 5B and Figure 7A). The upper part is much more voluminous (its diameter is about 50 µm) than the lower part, of which the diameter is about 5 µm, and gives rise to the cross nerves (Figure 7A) and semicircular nerves of the arm base (Figure 4C). The lower part of the main brachial nerve consists of a few neurites and forms a separate nerve tract, which I have called the **second main brachial nerve**.

**Cross nerves** originate from the upper portion of the main brachial nerve, extend into the extracellular matrix of the brachial arm (Figure 6B), and connect to the additional brachial nerve (Figure 4C and Figure 5B). According to the volume rendering, cross nerves are S-shaped (Figure 8A). A portion of the S-like cross nerve apparently extends under the epithelium of the food groove. Cross nerves are located regularly, and the distance between them is ≤15 µm (Figure 7A and Figure 8C). The thickness of cross nerves ranges from 5 to 10 µm (Figure 7A).

**The accessory brachial nerve** is located on the tentacular side of the food groove (Figure 6A). It is a thick nerve tract that contributes to the innervation of the tentacles (Figure 7B and Figure 8B). The accessory brachial nerve gives rise to thick nerves, which together form the **second accessory brachial nerve**. This nerve extends at the base of the lateral and abfrontal sides of the inner tentacles (Figure 6C). Some parts of this nerve extend into the epithelium of a ridge, which passes along the brachial arm (Figure 6C). The epithelial ridge forms the inner upper border of the food groove (Figure 6A). The accessory brachial nerve gives rise to the inner–outer nerves (Figure 8D), which extend into the extracellular matrix, and then into the outer side of the base of the lophophore, where they fuse with the radial brachial nerves.

The **lower brachial nerve** extends along the base of the brachial arm (Figure 6A). The lower brachial nerve is formed by three thick nerves and several thin neurites, which do not abut each other and do not form a prominent nerve tract (Figure 7C and Figure 8F). The lower brachial nerve connects to the radial brachial nerves, which skirt the base of lophophore arms and form a nerve net on the outer side of the brachial arm (Figure 7C,D). At the base of the abfrontal side of the outer tentacles, this nerve net forms a thin **additional lower nerve** (Figure 7D). This nerve has thick nodules, which are located between and at the bases of the outer tentacles (Figure 8E). According to volume rendering, frontal–abfrontal nerves, which extend between the outer tentacles and connect to the second accessory brachial nerve and the additional lower brachial nerve, contribute to the formation of these nodules (Figure 8E). The lower brachial nerve connects to the main brachial nerve via numerous thick, regularly extended nerves, which skirt the inner side of the base of each lophophore arm (Figure 4B,C and Figure 8F). These are the **nerves of the lophophore base**.

### 3.4. Innervation of Tentacles

The innervation of inner and outer tentacles is generally similar, but there are some differences (Figure 5B). Along each tentacle, three groups of nerves extend: one frontal, two lateral, and one abfrontal (Figure 3C,D). In the outer tentacles, there are two groups of lateroabfrontal nerves (Figure 3D). The inner tentacles lack lateroabfrontal nerves (Figure 3C). The frontal and laterofrontal nerves originate from the same lophophoral nerves in both the inner and outer tentacles, whereas the abfrontal nerves originate from different nerve tracts in the inner vs. the outer tentacles. In both the inner and outer tentacles, the frontal nerve arises from the accessory brachial nerve (Figure 5B and Figure 7B), and the laterofrontal nerves connect to the second accessory nerve. In the inner tentacles, the abfrontal nerve originates from the second accessory brachial nerve (Figure 5B). In the outer tentacles, the abfrontal and lateroabfrontal nerves extend from the additional lower brachial nerve (Figure 7D,E). The lateroabfrontal neurite bundles originate from the nodes, which are formed by the frontal–abfrontal brachial nerves (Figure 7E and Figure 8E). From each node, two nerves arise and extend to the adjacent outer tentacles (Figure 7E).

## 4. Discussion

### 4.1. Organization of the Central Nervous System in Brachiopods

The lophophore nervous system of brachiopods has been mostly studied by histological methods [29,30,33,34], and only a few species have been investigated by electron microscopy, immunocytochemistry, and confocal laser scanning microscopy [15,19,27,28,35,36,37]. Together, these studies indicate the general pattern of lophophore innervation.

The main neuronal elements are the ganglia, which are located near the mouth. In all rhynchonelliformes, there are two ganglia: the supraenteric is located above the mouth and the subenteric is located below the mouth. Neither of these “ganglia” is a true ganglion, i.e., they are represented only by a simple neuroepithelium or a stratified neuroepithelium [32]. Linguliform and craniiform adults have a subenteric ganglion but not a supraenteric ganglion [29,33,34]. Juveniles of the craniiform *Novocrania anomala*, however, have a prominent supraenteric ganglion, which is a bilaterally symmetrical structure with two lateral groups of perikarya that are connected to each other via a thick central nerve [19]. During development, the thick nerve becomes longer, and the distance between two lateral groups of perikarya increases. The failure to detect a supraenteric ganglion in *N. anomala* adults in the current study suggests that the thick central nerve of the juvenile supraenteric ganglion undergoes further extension and is transformed into the main brachial nerve of the adult.

In linguliform brachiopods, the swimming juvenile, which is usually called a “larva” [10], has a ganglion in the apical median tentacle above the mouth [38,39]. This ganglion, however, is not present in settled juveniles [38]. It follows that the apical ganglion in swimming stages of linguliform larvae can be compared with the apical organ of swimming larvae of other brachiopods [38,39,40] and with the apical organ of swimming bilaterally symmetrical ciliated larvae of other bilaterians [41,42]. In bilaterian larvae, the apical organ is important for planktonic life but becomes less important as metamorphosis continues [42,43]. In particular, the apical organ supplies the communication between growing larvae and adult animals [44]. At the same time, because settled linguliform juveniles lack the supratenteric ganglion, the apical ganglion of linguliform juveniles cannot be compared with the supraenteric ganglion of *N. anomala* juveniles. In other words, settled juvenile craniiforms and rhynchonelliforms have the supraenteric ganglion, whereas settled juvenile linguliforms lack the supraenteric ganglion (Table 1). Because species from two subphyla of recent brachiopods have the supraenteric ganglion, its presence probably represents the ancestral state, whereas its absence probably represents the apomorphic state.

The subenteric ganglion is represented by a thickened portion of the nerve that extends under the mouth in craniiforms and rhynchonelliforms [19,27,32]. In linguliforms, the subenteric ganglion is the prominent nerve nodule, which appears as the ventral ganglion in swimming juveniles and is maintained in adults [38,39]. Although the ultrastructure of the linguliform ventral ganglion has yet to be studied, immunocytochemistry suggests that it is organized as a typical ganglion with two groups of perikarya with a commissure between them [39]. Thus, the structure of the subenteric ganglion is similar in craniiforms and rhynchonelliforms but different in lingulifoms (Table 1).

In the brachiopods studied to date, there are three or four large brachial nerves: the main, accessory, second accessory, and lower nerve. All of these nerves (and some additional nerves) are present in *N. anomala*, but some are absent in rhynchonelliforms and linguliforms. More specifically, the accessory brachial nerve tends to be reduced in rhynchonelliforms; although it is present in *Hemithiris psittacea* [28], it does not contribute to the innervation of the tentacles, and it is completely absent in *Coptothyris grayi* [27]. In linguliforms, the second accessory nerve is absent; *Lingula anatina*, however, has groups of intertentacular perikarya located exactly between the rows of tentacles [15]. The second accessory brachial nerve apparently appeared in a brachiopod ancestor after the formation of the double row of tentacles; its formation has been reported in juveniles of *N. anomala* [18]. Moreover, the current study of adult *N. anomala* shows that the second accessory nerve is absent in the oral area, where tentacles from a single (not a double) row.

### 4.2. Innervation of the Tentacles in Brachiopods

In all studied brachiopods, the tentacles are innervated in a similar way (Table 1). In inner tentacles, there are four groups of nerves: one frontal, one abfrontal, and two laterofrontal. In outer tentacles, there are six groups of nerves: one frontal, one abfrontal, two laterofrontal, and two lateroabfrontal. The outer tentacles of *L. anatina* lack the lateroabfrontal nerves. The origin of nerves is similar in rhynchonelliforms and craniiforms, but differs in lingulliforms. This difference correlates with absence of the second accessory nerve in *L. anatina* (Table 1). It is interesting that the peritoneal neurites exhibit acetylated alpha-tubulin-like immunoreactivity in rhynchonelliforms but not in craniiforms or lingulliforms. At this moment, this specificity does not have an explanation. In order to understand the nature of this difference we need more data on immunoreactivity of nerve elements of tentacles in different brachiopod species from different groups. 

### 4.3. Evolution of the Lophophore

There are several ideas about how the lophophore evolved in lophophorates. According to the first idea [9,45,46,47], the lophophore of brachiopods evolved from an ancestral form whose small lophophore had a simple morphology to several advanced forms whose lophophores had complex morphologies. According to the second idea, which is based on morphological and paleontological data, the spirolophe, which is relatively complex, is the ancestral state [14]. The second idea suggests that the lophophore of recent brachiopods has evolved in two different ways: (i) simplification and (ii) complication. Both of these ways might correlate with changes in body size and the peculiarities of ontogeny. For some recent and extinct brachiopod species, the paedomorphic origin of the simple-shape lophophore (i.e., the taxolophe and trocholophe) is suggested [14]. 

In other lophophorates, i.e., phoronids and bryozoans, two different ways of lophophore evolution have been suggested [20,26]: (i) from a relatively large lophophore of complex morphology to a lophophore of simple morphology; and (ii) from a relatively large and complex lophophore to a lophophore of much more complex morphology. Because the homology of the lophophore has been established based on morphology, I suggest that the initial lophophore resembled a simple spirolophe (or a horseshoe-shaped lophophore) (Figure 9). In brachiopods, this ancestral type of lophophore changed due to the appearance of a double row of tentacles (Figure 9). It is important that the ancestral brachiopod lophophore had a double row of tentacles because the presence of a double row of tentacles led to the appearance of the second accessory brachial nerve. The formation of the second accessory brachial nerve due to the appearance of the outer tentacles was recently reported in the development of *N. anomala* [19].

## 5. Conclusions

Among the lophophores of all brachiopods, that of the *N. anomala* adult has the most complex nervous system, which consists of four large brachial nerves and several additional nerves. The presence of many additional nerves in the lophophore may correlate with the greatly developed muscles of the brachial arms. According to Table 1, at least nine characteristics (1a,3,6,7b,10,11,14,15,17) are similar in craniiforms and rhynchonelliforms, whereas only six features (1b,5,8,9,13,18) are shared by craniiforms and linguliforms. Some characteristics, which are used in the table, should be next tested with involvement of the correct outgroup in order to reveal characters’ polarity. At this moment, Table 1 allows the conclusion that the organization of the lophophore nervous system and the innervation of tentacles is generally similar between craniiforms and rhynchonelliforms but differs between the latter subphyla and the linguliforms. The similarities between craniiforms and rhynchonelliforms mostly concern the organization of the central nervous system, the structure of which is traditionally used for phylogenetical studies. These similarities and differences support the inference that craniiforms and rhynchonelliforms are more closely related to each other than to linguliforms, which is consistent with the “Calciata” hypothesis [13]. According to this hypothesis and the current results, the validity of the “Inarticulata” and the close relationship between Linguliformea and Craniiformea should be rejected.

## Figures and Tables

**Figure 1 biology-11-00406-f001:**
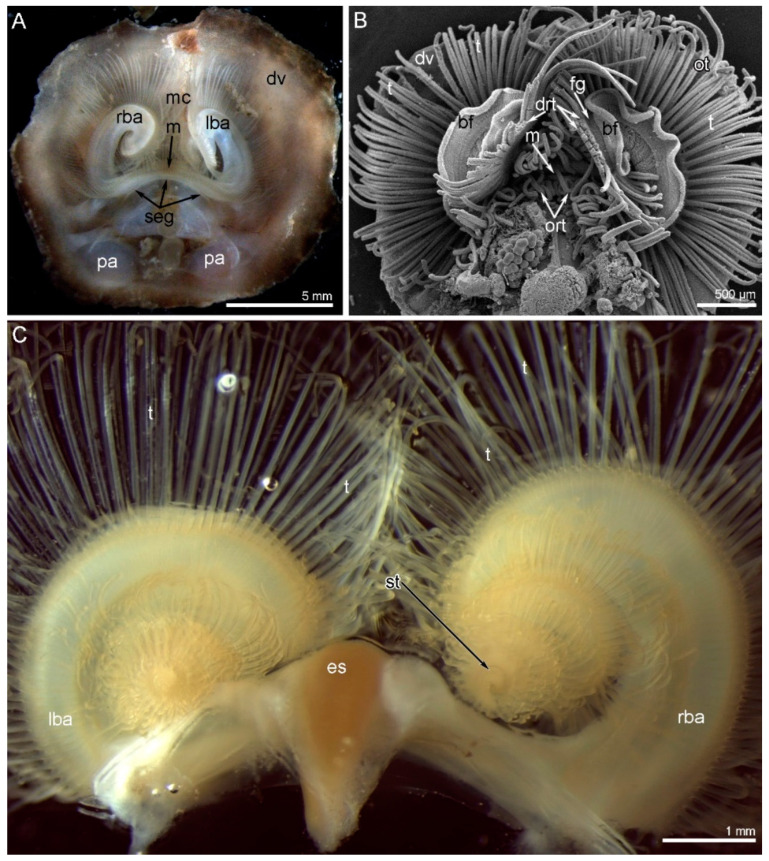
Organization of the lophophore in *Novocrania anomala* adults. (**A**) Photograph of a live animal viewed from the ventral side. (**B**) The same animal; SEM. (**C**) Photograph of separated spirolophe viewed from the dorsal side. The zones where new tentacles form are located on the distal end of each brachial arm. Abbreviations: bf—brachial fold; drt—double row of tentacles; dv—dorsal valve of the shell; es—esophagus; fg—food groove; lba—left brachial arm; m—mouth; mc—mantle cavity; ort—oral tentacles; ot—outer tentacle; pa—posterior adductor; rba—right brachial arm; seg—subenteric ganglion; st—zone of youngest tentacle; t—tentacle.

**Figure 2 biology-11-00406-f002:**
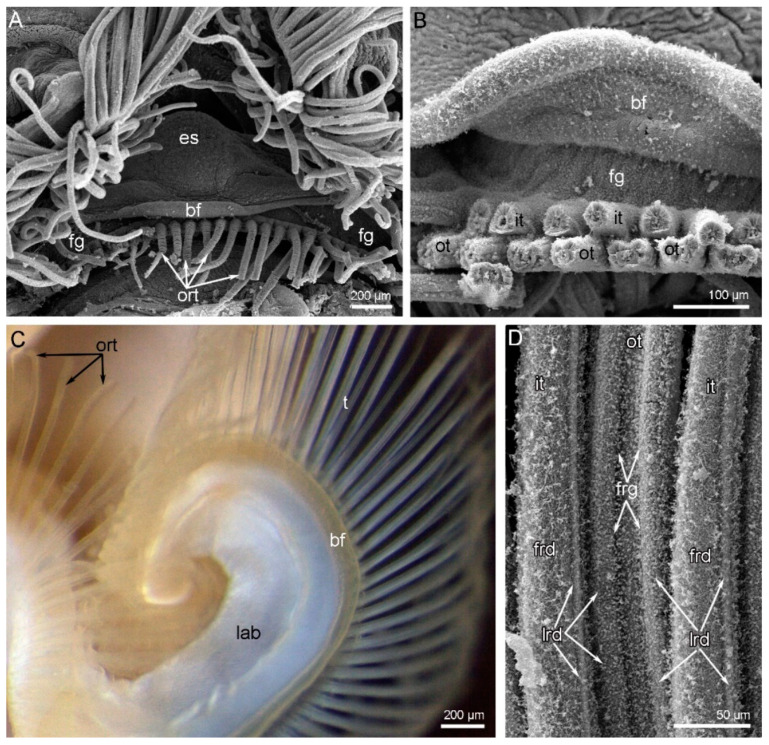
Details of the lophophore organization in *Novocrania anomala* adults. (**A**) The oral part of the lophophore; SEM. (**B**) A portion of the brachial arm: the brachial fold and double row of tentacles are visible; SEM. (**C**) The base of the lophophore arm, viewed from the ventral side; photograph of a live lophophore. (**D**) Fine morphology of tentacles: one outer and two inner tentacles are visible; SEM. Abbreviations: bf—brachial fold; es—esophagus; fg—food groove; frd—frontal ridge; frg—frontal groove; it—inner tentacle; lab—lophophore arm base; lrd—lateral ridge; ort—oral tentacles; ot—outer tentacle; t—tentacle.

**Figure 3 biology-11-00406-f003:**
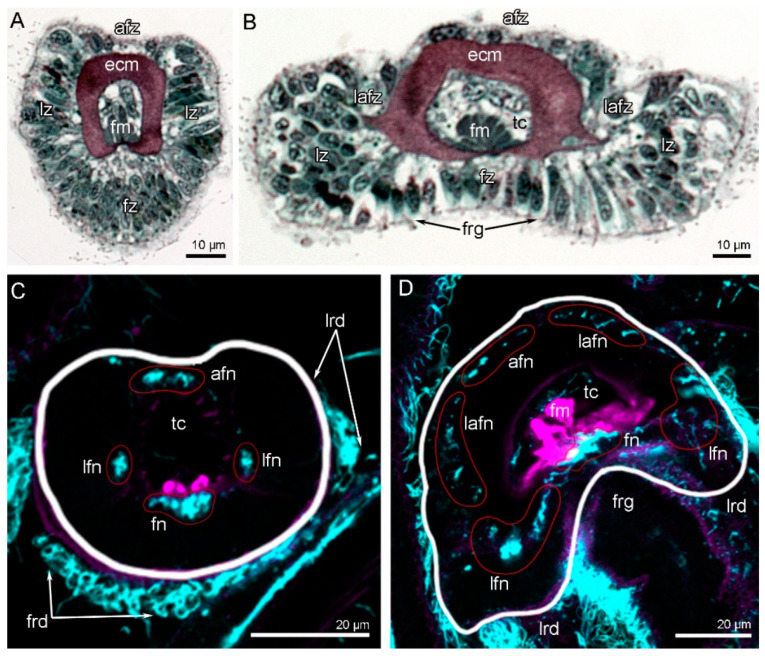
Tentacle organization and innervation in *Novocrania anomala* adults. Semi-thin cross-sections (**A**,**B**); Z-projections of two adjacent slides of the stack after immunostaining against acetylated alpha-tubulin (cyan) and staining with phalloidin (magenta) (**C**,**D**). White line shows the edge of the tentacles; red circles indicate certain tentacle nerves, which consist of several neurite bundles. (**A**) Inner tentacle. (**B**) Outer tentacle. (**C**) Z-projection of cross-section of the inner tentacle. (**D**) Z-projection of cross-section of the outer tentacle. Abbreviations: afn—abfrontal tentacle nerve; afz—abfrontal zone; ecm—extracellular matrix; fm—frontal muscle; fn—frontal tentacle nerve; frd—frontal ciliated ridge; frg—frontal groove; fz—frontal zone; lafn—lateroabfrontal tentacle nerve; lafz—lateroabfrontal zone; lfn –laterofrontal tentacle nerve; lrd—lateral ciliated ridge; lz—lateral zone; tc—tentacle coelom.

**Figure 4 biology-11-00406-f004:**
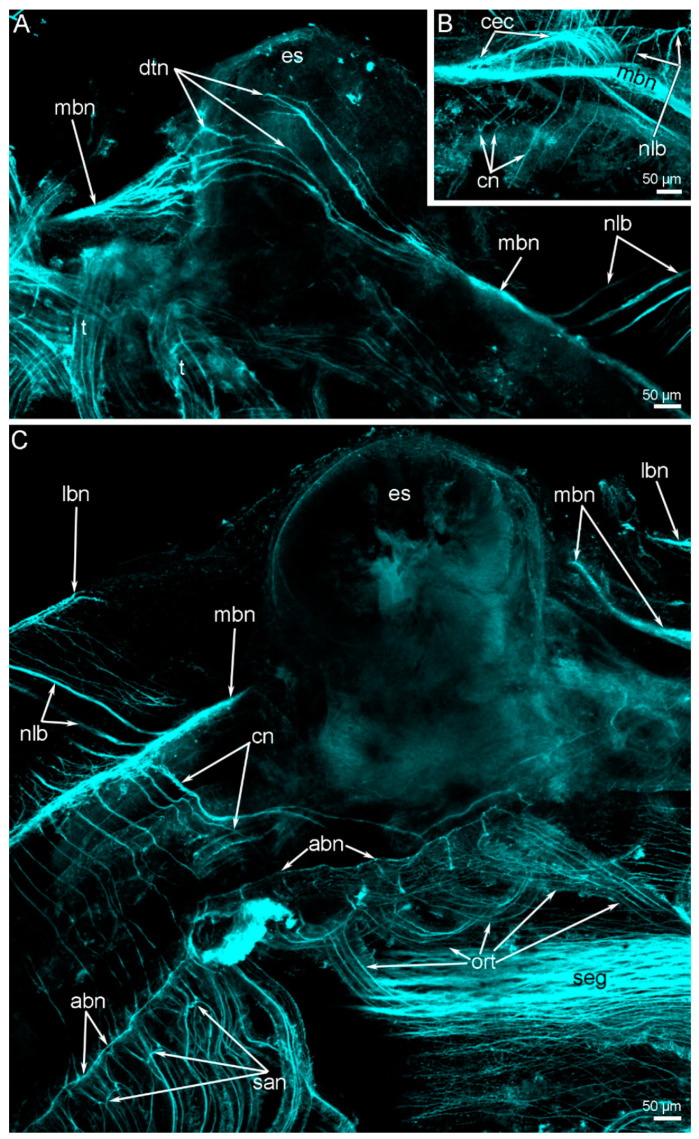
Organization of the nervous system in the oral region of *Novocrania anomala* adults. Z-projections after immunostaining against acetylated alpha-tubulin (cyan). (**A**) The dorsal view of the esophagus: supraenteric ganglion is absent; main brachial nerves of two arms are connected via dorsal thin neurites. (**B**) The base of the lophophore arm: circumenteric connective extends from the main brachial nerve. (**C**) Ventral view of the esophagus and oral area with the single row of oral tentacles. Abbreviations: abn—accessory brachial nerve; cec—circumenteric connective; cn—cross nerve; dtn—dorsal thin neurites; es—esophagus; lbn—lower brachial nerve; mbn—main brachial nerve; nlb—nerve of the lophophore base; ort—oral tentacles; san—second accessory brachial nerve; seg—subenteric ganglion.

**Figure 5 biology-11-00406-f005:**
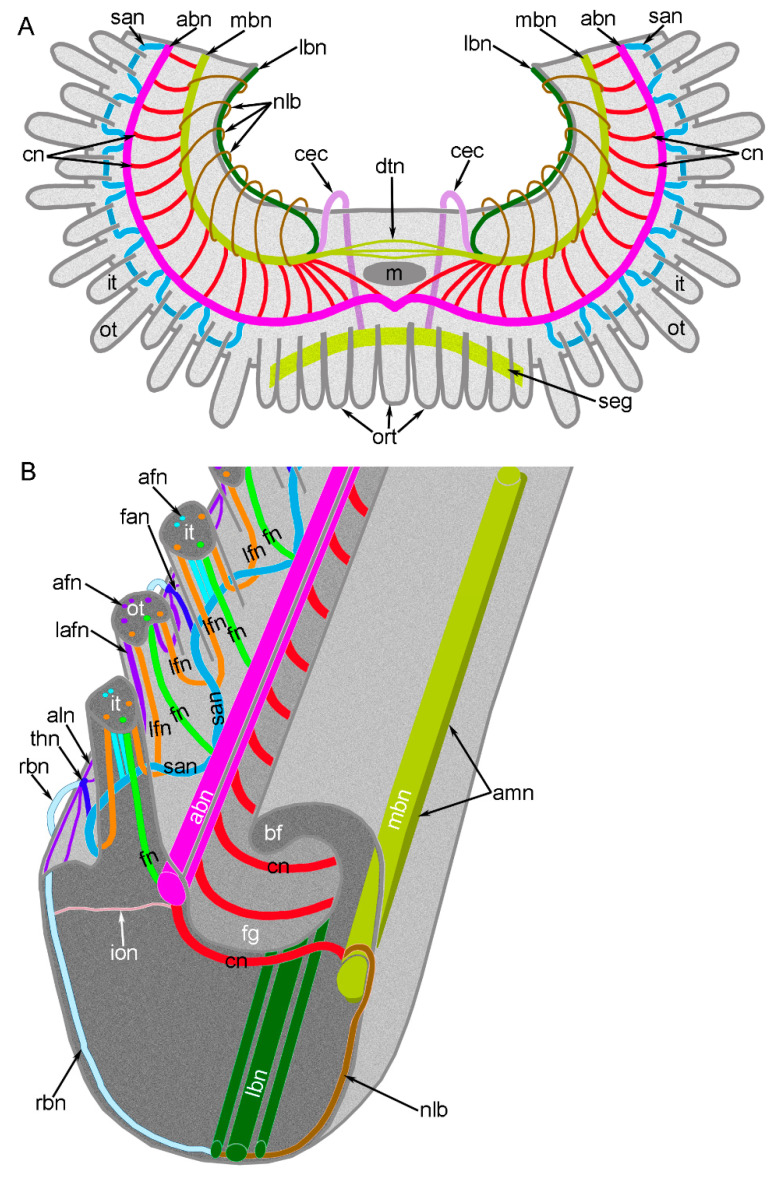
Schemes of innervation of the lophophore and tentacles in *Novocrania anomala* adults. (**A**) General scheme of the lophophore nerve elements. (**B**) A portion of the brachial arm with double row of tentacles. Abbreviations: afn—abfrontal tentacle nerve; abn—accessory brachial nerve; aln—additional lower brachial nerve; amn—additional main brachial nerve; bf—brachial fold; cec—circumenteric connective; cn—cross nerve; dtn—dorsal thin neurites; fan—frontal–abfrontal nerve; fg—food groove; fn—frontal tentacle nerve; ion—inner–outer nerve; it—inner tentacle; lafn—lateroabfrontal tentacle nerve; lbn—lower brachial nerve; lfn—laterofrontal tentacle nerve; m—mouth; mbn—main brachial nerve; nlb—nerve of the lophophore base; ort—oral tentacles; ot—outer tentacle; rbn—radial brachial nerve; san—second accessory nerve; seg—subenteric ganglion; thn—thick nodule.

**Figure 6 biology-11-00406-f006:**
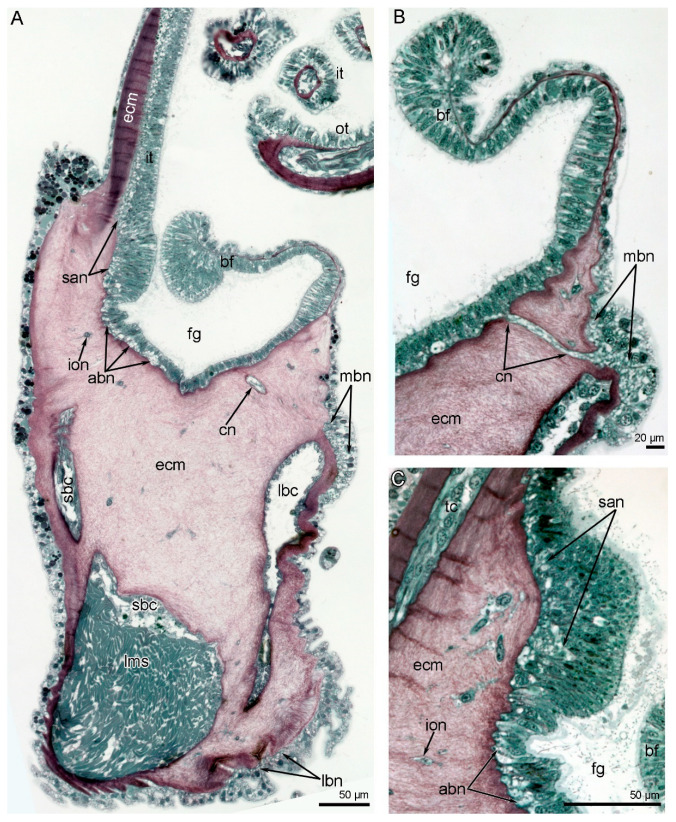
Brachial nerves in the semi-thin cross-sections of the lophophore arm in *Novocrania anomala* adults. (**A**) General view of a cross-section of the arm: the location of the brachial nerves is evident. (**B**) The main brachial nerve giving rise to the cross nerve. (**C**) Inner side of the base of the double row of tentacles: accessory and second accessory brachial nerves are visible. Abbreviations: abn—accessory brachial nerve; bf—brachial fold; cn—cross nerve; ecm—extracellular matrix; fg—food groove; ion—inner–outer nerve; it—inner tentacle; lbc—large brachial coelomic canal; lbn—lower brachial nerve; lms—large muscle of small coelomic canal; mbn—main brachial nerve; ot—outer tentacle; san—second accessory brachial nerve; sbc—small brachial coelomic canal; tc—tentacle coelom.

**Figure 7 biology-11-00406-f007:**
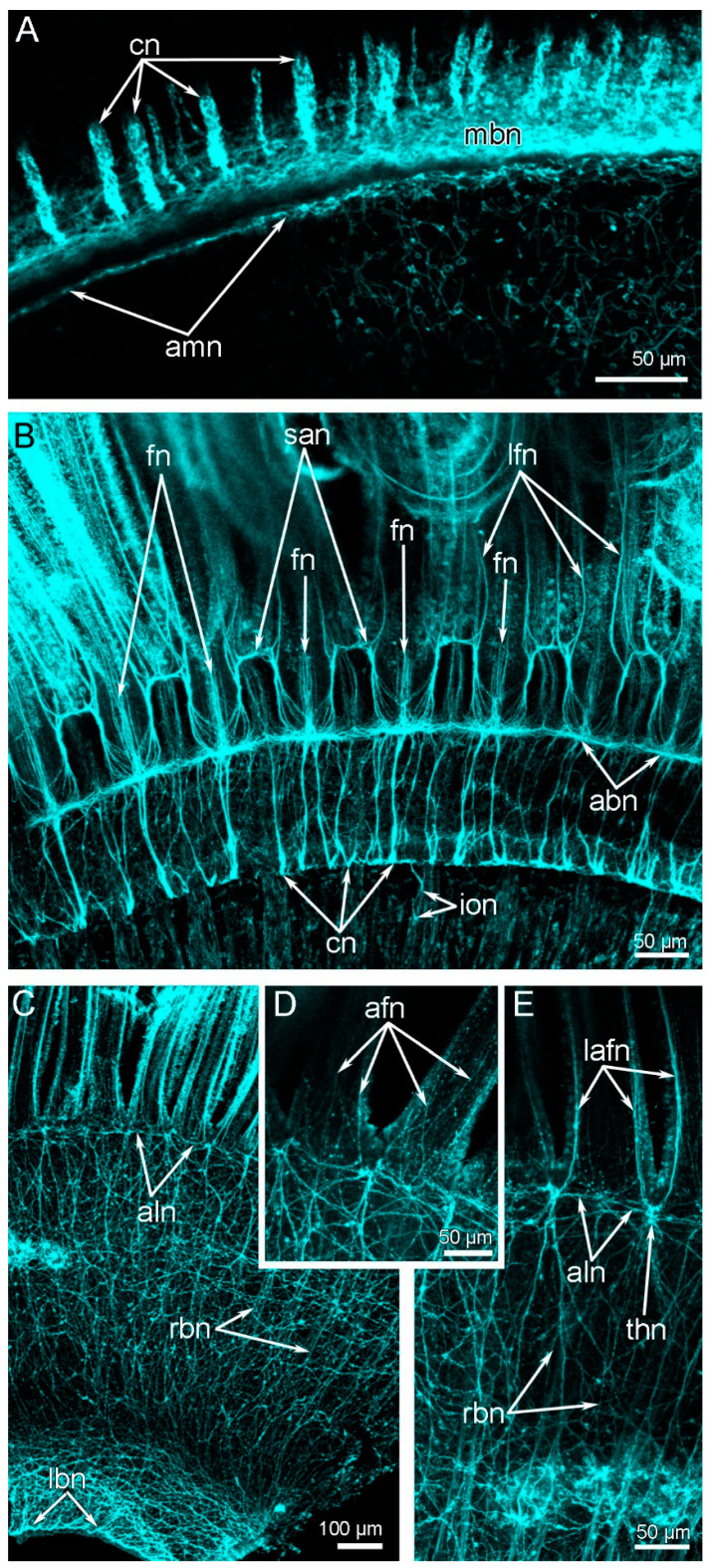
Brachial nerves and the innervation of tentacles in *Novocrania anomala* adults. Z-projections after immunostaining against acetylated alpha-tubulin (cyan). (**A**) The main brachial nerve and its additional part—the additional main brachial nerve. (**B**) The accessory and second accessory brachial nerves; frontal and latero-frontal tentacle nerves of both inner and outer tentacles originate from the second accessory nerve. (**C**) Outer side of the lophophore arm: the lower and additional lower nerve are visible. (**D**) The base of two outer tentacles. (**E)** The base of the outer tentacles: lateroabfrontal tentacle nerves originate from the thick nodules of the additional lower brachial nerve. Abbreviations: afn—abfrontal tentacle nerve; abn—accessory brachial nerve; aln—additional lower brachial nerve; amn—additional main brachial nerve; cn—cross nerve; fn—frontal tentacle nerve; ion—inner–outer nerve; lafn—lateroabfrontal tentacle nerve; lbn—lower brachial nerve; lfn—laterofrontal tentacle nerve; mbn—main brachial nerve; rbn—radial brachial nerve; san—second accessory nerve; thn—thick nodule.

**Figure 8 biology-11-00406-f008:**
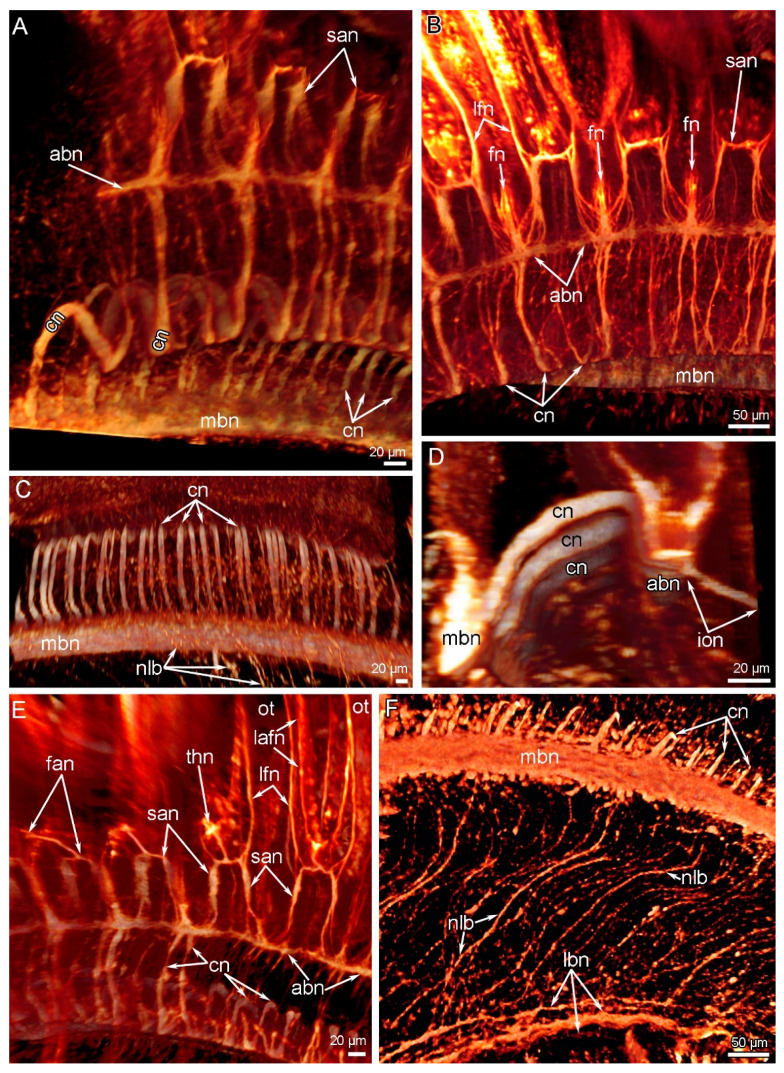
Volume rendering of the nerve elements of a part of the lophophore arm in *Novocrania anomala* adults. Reconstruction after immunostaining against acetylated alpha-tubulin. (**A**) Shape of cross nerves and their connection to the accessory brachial nerve. (**B**) Accessory and second accessory brachial nerves. (**C**) A portion of cross nerves, which extend along the bottom of the food groove. (**D**) Inner–outer nerve originating from the accessory brachial nerve. (**E**) Innervation of tentacles: frontal–abfrontal nerve and thick nodule are visible. (**F**) Nerves of the lophophore base that connect the main and lower brachial nerves. Abbreviations: abn—accessory brachial nerve; cn—cross nerve; fan—frontal–abfrontal nerve; fn—frontal tentacle nerve; ion—inner–outer nerve; lafn—lateroabfrontal tentacle nerve; lbn—lower brachial nerve; lfn—laterofrontal tentacle nerve; lbn—lower brachial nerve; mbn—main brachial nerve; nlb—nerves of the lophophore arm base; ot—outer tentacle; san—second accessory nerve; thn—thick nodule.

**Figure 9 biology-11-00406-f009:**
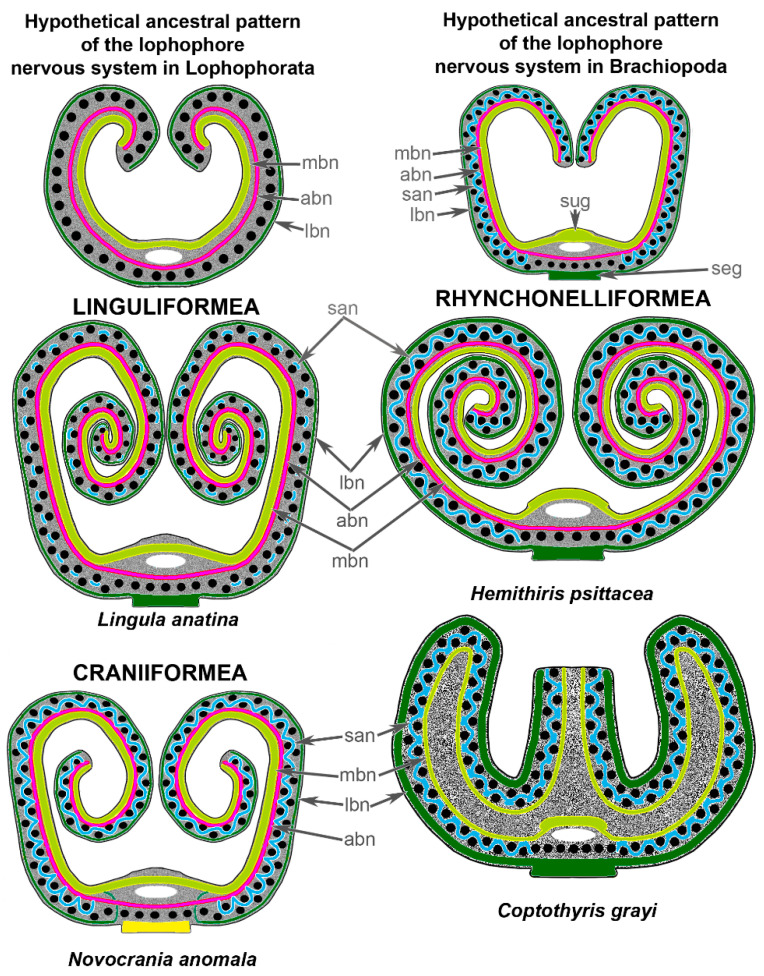
Scheme of the lophophore nervous system in brachiopods from all three subphyla. The reconstruction is based on previous [15,27,28] and recent studies. The black circles indicate tentacles. Colors indicate different nerve elements of the lophophore: yellow–green—main brachial nerve and the supraenteric ganglion; pink—accessory brachial nerve; blue—second accessory brachial nerve; dark green—lower brachial nerve and the subenteric ganglion. Abbreviations: abn—accessory brachial nerve; lbn—lower brachial nerve; mbn—main brachial nerve; san—second accessory brachial nerve; seg—subenteric ganglion; sug—supraenteric ganglion.

**Table 1 biology-11-00406-t001:** Nerve elements of the lophophore in the three subphyla of brachiopods.

Number	Feature	Rhynchonelliformea [27,28]	Craniiformea [17,29], herein	Linguliformea [15,33]
1a	supraenteric ganglionin juvenile	+	+	―
1b	supraenteric ganglionin adult	+	―	―
2	subenteric ganglion	+	+	+
3	organization of subenteric ganglion	a portion of the nerve	a portion of the nerve	ganglion
4	main brachial nerve	+	+	+
5	accessory brachial nerve	―	+	+
6	second accessory nerve	+	+	―
7a	lower brachial nerve structure: nerve net	+	+	+
7b	lower brachial nerve structure: solid nerve	+	+	―
8	origin of the lower brachial nerve	ganglionic	circumenteric	circumenteric
Innervation of inner tentacles				
9	frontal nerve	from the cross nerves	from accessory brachial nerve	from accessory brachial nerve
10	laterofrontal nerves	from second accessory nerve	from second accessory nerve	from intertentacular nerves
11	abfrontal nerve	from second accessory nerve	from second accessory nerve	from intertentacular perikarya
12	lateroabfrontal nerves	―	―	―
13	immunoreactive peritoneal neurites	+	―	―
Innervation of outer tentacles				
14	frontal nerve	from second accessory nerve	from second accessory nerve	from intertentacular nerves
15	laterofrontal nerves	from second accessory nerve	from second accessory nerve	from intertentacular nerves
16	abfrontal nerve	from lower brachial nerve	from lower brachial nerve	from lower brachial nerve
17	lateroabfrontal nerves	from lower brachial nerve	from lower brachial nerve	―
18	immunoreactive peritoneal neurites	+	―	―

“―” indicates the absence of the structure. “+” indicates the presence of the structure.

## Data Availability

The data sets analyzed during this study are available from the author upon request.

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
