# Peer review of "First Modern Data on the Lophophore Nervous System in Adult Novocrania anomala and a Current Assessment of Brachiopod Phylogeny"

_biology, 2022, doi:10.3390/biology11030406_

Round 1
Reviewer 1 Report
The manuscript “First modern data on the lophophore nervous system in adult Novocrania anomala and a current assessment of brachiopod phylogeny” by Elena Temereva presents new insights on the neuroanatomy of a rather understudied taxon the brachiopods. By application of modern techniques, the author provides a nice descriptive paper with 9 figures and one table. The study is well done and sound, nevertheless there are many smaller flaws, especially in the description of methods and presentation of figures. In general, I have no crucial concerns regarding the integrity of this contribution and think that it is well suited for publication in this journal. Nevertheless, I have some major questions and a huge number of minor issues, which should be addressed in a revised version before publication and which will help to make it a much stronger piece.
Major comments:
- In first place, the manuscript compares the neuroanatomy of the three subphyla within Brachiopoda (explicitly stated in the title). However, especially in the discussion, the author also touches the relationship of all lophophorates based on neuroanatomy. For the latter, too less information is given, especially in the introduction. For example, which are the main characteristics of the nervous system in lophophorates? Which characteristics do support the homology of the lophophore? Which are the arguments against such a homology? And in the discussion, the author should clearly address in which way the new data fit in these ideas: Is the lophophore a homologue structure or not? Also, how are the three taxa of lophophorates related to each other, based on current concepts of phylogeny? I think the author needs to come to a decision: Do you want to focus on brachiopod phylogeny only, or do you want to include the lophophorate debate as well? If the later applies, you need to extend the given information and background (and should also think about the extension of Table 1 for remaining lophophorates).
- Related to the first point: It might be a good idea to show a figure with hyoptheses on phylogenetic relationships of brachiopods and lophophorates, maybe associated with supporting neuroanatomical evidence. Along these lines, I cannot follow the conclusion, that Calciata-hypothesis (supported by this contribution) is supported by the fact that Cancriiformea is the youngest group of brachiopods. If Cancrifformea and Rhynchonelliformea are sister groups, how could one be proposed to be younger than the other?
- Further, one of the main reasons, why this study has been performed is the fact, that juvenile and adult nervous systems differ. This is not clearly explained in the introduction. Please address these differences already here, the reader will be able to follow your story a lot better.
- The figures need some modifications. Especially the orientation (anterior, posterior, etc) and direction of sectioning (cross, sagittal or horizontal sections) of figures needs to be added. That will allow scientists not that familiar with brachiopod anatomy to follow your explanations. Furthermore, see detailed comments below.
- One question concerning acetylated-alpha-tubulin staining: The author states that no labelling with this antibody occurs in peritoneal neurites in inner and outer tentacles in Craniiformea and Linguliformea. I do not understand, how it is possible that some neurites are labeled (as you show in your figures) and others not. Neurites should have tubulin in their cytoskeleton. Are you sure that this cannot be a reason of methodology, e.g. penetration problems?
Line-by-line comments:
Simple Summary:
Line 15: delete doubled full stop.
Abstract:
Line 19: Change “Rhinchonelliformea” to “Rhynchonelliformea”
Line 27: change to “…organization of the subenteric ganglion…”
Introduction:
Line 42: Change “atriculate” to “articulate”
Line 48: “The lophophore is tentacular organ that is present all lophophorates” should read “The lophophore is a tentacular organ that is present in all lophophorates”.
Material and Methods:
Line 83: The heading “Microscopy” is misleading, as this chapter describes also the methods for fixation and labelling.
In general, the methods are described too brief, for example:
- In Figure 3 a phalloidin staining is shown. However, nothing is mentioned about phalloidin staining in the Material and Methods!
- Line 83f.: In which solution is glutaraldehyde and osmium tetroxide diluted? Times of fixation?
- Line 84: steps of ethanol series and incubation times?
- Line 86: Please give the thickness of sections?
- Line 88: Give more information about the methylene blue staining, how has that been done (e.g., incubation times etc). What about the mounting medium for microscopy?
- Line 91: temperature for fixation?
- Line 97: Dilution medium for PFA? Temperature and times for PFA incubation?
- Blocking: incubation time? By the way: Why have you used normal donkey serum? For better efficiency, you should use the serum from the species, where the secondary antibody comes from (in your case normal goat serum).
- Line 102: change “mouse anti-a-tubulin” to “mouse anti-acetylated-a-tubulin”.
- Line 103: secondary antibody diluted in?
Line 115: “the terminology…devoted to the brachiopod nervous system is often inconsistent…”. One important and very confusing term used in this manuscript is the term “ganglion”. The term is first used in line 152: “The subenteric ganglion is represented by a thick nerve”. This sentence is an oxymoron! The author clarifies this in the discussion, but this is far to late. It should be touched before using the term ganglion - maybe here in the terminology section. Furthermore, the term “thick nodule” is mentioned several times and in Table 1 defined as a “true ganglion”. Could you please comment on that? Is a thick nodule a ganglion or not? If so, please clarify here.
Results:
- line 136: “double row tentacles” should read “double row of tentacles”.
- Lines 137/140: The author states that inner and outer tentacles differ in morphology. Please give details here, which are the morphological differences?
- Line 143: “is opposite the frontal zone” should read “is opposite to the frontal zone”.
- Line 182f.: Change “where they connect the radial brachial nerves” to “where they connect to the radial brachial nerves” or alternatively “where they fuse with the radial brachial nerves”.
Discussion:
- Line 312: “The main nerve elements are the ganglia”. That is wrong, a ganglion cannot be part of a nerve (see also my comment concerning terminology issues above).
- Line 316: Add closed square bracket and full stop after “neuroepithelium [29”.
- Line 326: is that a true ganglion or a nerve?
- Line 342: “adult” should be plural.
- Line 375: Change “…is the ancestral [6]. Second idea…” to “…is the ancestral state [6]. The second idea…”.
- Lines 375-379: This paragraph lacks some references.
- Line 381: delete “and”
Fig. 2:
Line 223: What exactly means “inner side”? This is not a very precise anatomical description.
Explanation of the abbreviation “t” is missing in the figure legend.
Fig. 3:
- Red circles are hardly visible, especially in combination with magenta staining. Please chose better color combination.
- Please give information about the thickness covered by the z-stacks and the distance between adjacent images taken by CLSM (also applicable for other CLSM panels).
- Abbreviations should be in alphabetical order
- Line 234f.: abbreviation “c2” is not labeled in the figure
- Abbreviations “lrd” and “frd” are not explained in the figure legend.
Fig. 5:
- Abbreviations: “ln”, “m”, “bf”, and “fg” are not explained in the figure legend.
- I am wondering, if the extensive and kind of confusing labelling with abbreviations could be replaced by a color code table insight of the figure (see also figure 9)?
Fig. 6:
- Please add scale bars!
- Line 264: “second accessory nerves” should read “second accessory brachial nerves”?
Fig. 7:
- “mbn” in A is difficult to read, I recommend white letters with black outline.
- Line 278: replace “tentacler” by “tentacle”
- Line 281: delete explanation for “lbn”, it is doubled (compare line 280).
Fig. 8:
- Please add scale bars!
- Black lettering in D is sometimes hardly visible (e.g. “lbn”, or lower “cn”), white with black outline might work better.
- Delete second explanation of “lbn”, it is doubled.
- Line 294: abbreviation “rbn” is not labeled in the figure
Fig. 9:
- A color-code plate could be helpful to understand the figure without the need for checking abbreviations.
- The single schemes seem kind of crowded, more space between them would be beneficial.
- The paper investigated the neuroanatomy in Craniiformea. Thus, the other schemes must be based on other literature – these papers should be cited.
Table 1:
- Heading should read “…in the three subphyla…”
- Supraenteric ganglion: For clarity, I recommend adding “+ in juvenile, + in adult” for the subphylum Rhynchonelliformea.
- “a structure of subenteric ganglion” sounds strange to me. Following the states for the subphyla this point differentiates between the appearance of the subenteric ganglion. So what about Neuroanatomical appearance of…” or “Organization of…”?
- Same line: “a portion of the nerve extending…”. This wording implies reference to a specific nerve, which one? If not, please rephrase.
- “direction of the lower brachial nerve” and “direction of the cross nerves”: The term direction is misleading. I think you mean origin or projection?
- Heading “Other nerves of the lophophore”: As you also write about perikaryal, this heading might be better phrased “Other neuronal structures of the lophophore”?
- The author states positive FMRF-amide-like IR in certain perikarya in Linguliformea. My question is: IS FMRF absent in the other subphyla or has it just not been applied to them? That should be indicted, because it makes a big difference for the interpretation of the table.
- “Lateroabfrontal tentacle nerve” is absent in all three subphyla. It is not clear for the reader, why this information is important. Probably, because it is present in other lophophorates? But then, the author should extend the table by information on bryozoans and phoronids.
Author contributions, lines 410f.: I am wondering who performed the experiments and microscopy?? :)
References: I have not checked the references into any detail, but I realized that year of publication is missing in the references 9-11.
Author Response
Dear Reviewer,
Thank you very much for time that you have devoted to the manuscript and for your useful suggestions and comments. I have addressed all your comments. Please find below my point-by-point answer.
Beast regards,
Elena Temereva
RESPONSE TO REVIEWER 1:
- However, especially in the discussion, the author also touches the relationship of all lophophorates based on neuroanatomy. For the latter, too less information is given, especially in the introduction. For example, which are the main characteristics of the nervous system in lophophorates? Which characteristics do support the homology of the lophophore? Which are the arguments against such a homology? And in the discussion, the author should clearly address in which way the new data fit in these ideas: Is the lophophore a homologue structure or not? Also, how are the three taxa of lophophorates related to each other, based on current concepts of phylogeny? I think the author needs to come to a decision: Do you want to focus on brachiopod phylogeny only, or do you want to include the lophophorate debate as well? If the later applies, you need to extend the given information and background (and should also think about the extension of Table 1 for remaining lophophorates).
ANSWER: Actually, I did not want to touch relationship of all lophophorates. I have just mentioned that the problem of the evolution of the lophophore is known in other lophophorates.
- Related to the first point: It might be a good idea to show a figure with hyoptheses on phylogenetic relationships of brachiopods and lophophorates, maybe associated with supporting neuroanatomical evidence. Along these lines, I cannot follow the conclusion, that Calciata-hypothesis (supported by this contribution) is supported by the fact that Cancriiformea is the youngest group of brachiopods. If Cancrifformea and Rhynchonelliformea are sister groups, how could one be proposed to be younger than the other?
ANSWER: I have deleted the sentence about craniiformes as youngest brachiopods
- Further, one of the main reasons, why this study has been performed is the fact, that juvenile and adult nervous systems differ. This is not clearly explained in the introduction. Please address these differences already here, the reader will be able to follow your story a lot better.
ANSWER: I have added information
- The figures need some modifications. Especially the orientation (anterior, posterior, etc) and direction of sectioning (cross, sagittal or horizontal sections) of figures needs to be added. That will allow scientists not that familiar with brachiopod anatomy to follow your explanations. Furthermore, see detailed comments below.
ANSWER: I have made all corrections, which are suggested by Reviewer.
- One question concerning acetylated-alpha-tubulin staining: The author states that no labelling with this antibody occurs in peritoneal neurites in inner and outer tentacles in Craniiformea and Linguliformea. I do not understand, how it is possible that some neurites are labeled (as you show in your figures) and others not. Neurites should have tubulin in their cytoskeleton. Are you sure that this cannot be a reason of methodology, e.g. penetration problems?
ANSWER: I do not have the answer. The lophophores of craniiform Novocranial anomala and rhynchonelliform Hemithiris psittacea are similar in morphology and structure. The tentacles in both species are also similar in diameter and in structure. To study both species I have used similar protocol and the same chemical agents. However, in rhynchonelliform Hemithiris psittacea, peritoneal neurites exhibit immunoreactivity, whereas in N. anomala these neurites lack such reactivity. The same concerns phoronids and bryozoans: they also have peritoneal neurites, which do not demonstrate tubulin-like immunoreactivity. At this moment, I do not have enough data to explain this situation.
MINOR COMMENTS:
Line 15: delete doubled full stop.---DONE
Line 19: Change “Rhinchonelliformea” to “Rhynchonelliformea” --------- DONE
Line 27: change to “…organization of the subenteric ganglion…” ------DONE
Line 42: Change “atriculate” to “articulate” ------DONE
Line 48: “The lophophore is tentacular organ that is present all lophophorates” should read “The lophophore is a tentacular organ that is present in all lophophorates”. ---------------DONE
Line 83: The heading “Microscopy” is misleading, as this chapter describes also the methods for fixation and labelling. --------Changed to FIXATION AND MICROSCOPY
In Figure 3 a phalloidin staining is shown. However, nothing is mentioned about phalloidin staining in the Material and Methods! ----------Description is added
Line 83f.: In which solution is glutaraldehyde and osmium tetroxide diluted? Times of fixation? ------------- Information is added
Line 84: steps of ethanol series and incubation times? ------------- Information is added
Line 86: Please give the thickness of sections? ------------- Information is added
Line 88: Give more information about the methylene blue staining, how has that been done (e.g., incubation times etc). What about the mounting medium for microscopy? ------------- Information is added
Line 91: temperature for fixation? ------------- Information is added
Line 97: Dilution medium for PFA? Temperature and times for PFA incubation? ------------- Information is added
Blocking: incubation time? By the way: Why have you used normal donkey serum? For better efficiency, you should use the serum from the species, where the secondary antibody comes from (in your case normal goat serum). ------------- Thank you. You are right. It is mistake. I have used exactly normal goat serum. The correction is done.
Line 102: change “mouse anti-a-tubulin” to “mouse anti-acetylated-a-tubulin”. ------------- DONE
Line 103: secondary antibody diluted in? ------------- Information is added
Line 115: “the terminology…devoted to the brachiopod nervous system is often inconsistent…”. One important and very confusing term used in this manuscript is the term “ganglion”. The term is first used in line 152: “The subenteric ganglion is represented by a thick nerve”. This sentence is an oxymoron! The author clarifies this in the discussion, but this is far to late. It should be touched before using the term ganglion - maybe here in the terminology section. Furthermore, the term “thick nodule” is mentioned several times and in Table 1 defined as a “true ganglion”. Could you please comment on that? Is a thick nodule a ganglion or not? If so, please clarify here. ------------- Information is added
********************************
line 136: “double row tentacles” should read “double row of tentacles”. ------------- DONE
Lines 137/140: The author states that inner and outer tentacles differ in morphology. Please give details here, which are the morphological differences? ------------- DONE
Line 143: “is opposite the frontal zone” should read “is opposite to the frontal zone”. ------------- DONE
Line 182f.: Change “where they connect the radial brachial nerves” to “where they connect to the radial brachial nerves” or alternatively “where they fuse with the radial brachial nerves”. ------------- DONE
*******************************
Line 312: “The main nerve elements are the ganglia”. That is wrong, a ganglion cannot be part of a nerve (see also my comment concerning terminology issues above).----------------------- I did not understand this. I did not stay that ganglion is a part of nerve. Nerve elements and nerves are two big difference.
Line 316: Add closed square bracket and full stop after “neuroepithelium [29”. ------------- DONE
Line 326: is that a true ganglion or a nerve? -----------------------In larva, it is ganglion
Line 342: “adult” should be plural. ------------- DONE
Line 375: Change “…is the ancestral [6]. Second idea…” to “…is the ancestral state [6]. The second idea…”. ------------- DONE
Lines 375-379: This paragraph lacks some references. ----------------The reference is added
Line 381: delete “and”------------DONE
Fig. 2:
Line 223: What exactly means “inner side”? This is not a very precise anatomical description.------------Yes, it is ventral side. Change is done
Explanation of the abbreviation “t” is missing in the figure legend. ------------Explanation is added
Fig. 3:
Red circles are hardly visible, especially in combination with magenta staining. Please chose better color combination.----------------- I prefer this color, it does not draw reader attention from the main structures
Please give information about the thickness covered by the z-stacks and the distance between adjacent images taken by CLSM (also applicable for other CLSM panels).------------------------ DONE
Abbreviations should be in alphabetical order--------------DONE
Line 234f.: abbreviation “c2” is not labeled in the figure-------------------Deleted
Abbreviations “lrd” and “frd” are not explained in the figure legend.---------Added
Fig. 5:
Abbreviations: “ln”, “m”, “bf”, and “fg” are not explained in the figure legend.-----------------Added
I am wondering, if the extensive and kind of confusing labelling with abbreviations could be replaced by a color code table insight of the figure (see also figure 9)? ------------- I prefer to use different colors for different structures and also use abbreviations.
Fig. 6:
Please add scale bars!---------DONE
Line 264: “second accessory nerves” should read “second accessory brachial nerves”?-------------DONE
Fig. 7:
“mbn” in A is difficult to read, I recommend white letters with black outline.----------DONE
Line 278: replace “tentacler” by “tentacle” .----------DONE
Line 281: delete explanation for “lbn”, it is doubled (compare line 280).----------DONE
Fig. 8:
Please add scale bars!----------DONE
Black lettering in D is sometimes hardly visible (e.g. “lbn”, or lower “cn”), white with black outline might work better.--------DONE
Delete second explanation of “lbn”, it is doubled.------------Deleted
Line 294: abbreviation “rbn” is not labeled in the figure ------------Deleted
Fig. 9:
A color-code plate could be helpful to understand the figure without the need for checking abbreviations.--------------Color code is added
The single schemes seem kind of crowded, more space between them would be beneficial.
The paper investigated the neuroanatomy in Craniiformea. Thus, the other schemes must be based on other literature – these papers should be cited.------------DONE
************************************************************
Table 1:
- Heading should read “…in the three subphyla…”----------DONE
- Supraenteric ganglion: For clarity, I recommend adding “+ in juvenile, + in adult” for the subphylum Rhynchonelliformea.----DONE
- “a structure of subenteric ganglion” sounds strange to me. Following the states for the subphyla this point differentiates between the appearance of the subenteric ganglion. So what about Neuroanatomical appearance of…” or “Organization of…”?----------DONE
- Same line: “a portion of the nerve extending…”. This wording implies reference to a specific nerve, which one? If not, please rephrase.-----------DONE
- “direction of the lower brachial nerve” and “direction of the cross nerves”: The term direction is misleading. I think you mean origin or projection?-------------DONE
- Heading “Other nerves of the lophophore”: As you also write about perikaryal, this heading might be better phrased “Other neuronal structures of the lophophore”?-----------This part is deleted accordingly to recommendation of Reviewer 2
- The author states positive FMRF-amide-like IR in certain perikarya in Linguliformea. My question is: IS FMRF absent in the other subphyla or has it just not been applied to them? That should be indicted, because it makes a big difference for the interpretation of the table.------------------ This part is deleted accordingly to recommendation of Reviewer 2
- “Lateroabfrontal tentacle nerve” is absent in all three subphyla. It is not clear for the reader, why this information is important. Probably, because it is present in other lophophorates? But then, the author should extend the table by information on bryozoans and phoronids.--------------- This information is important because these nerves are present in the outer tentacles
Author contributions, lines 410f.: I am wondering who performed the experiments and microscopy?? :)--------------- Corrected
References: I have not checked the references into any detail, but I realized that year of publication is missing in the references 9-11.-------Corrected

Reviewer 2 Report
Please find my comments in the pdf attached

Author Response
Dear Reviewer,
Thank you very much for time that you have devoted to the manuscript and for your useful suggestions and comments. I have addressed all your comments. Please find below my point-by-point answer.
Beast regards,
Elena Temereva
RESPONSE TO REVIEWER 2:
- Because of this, I re-wrote the table (seebelow).
ANSWER: I am extremely grateful to Reviewer 2 for this great suggestion. I have changed the Table accordingly to Reviewer comments. I have also took into account Reviewer’s recommendations about non comparable data and deleted some features from the Table.
- This table shows that 8 characters are identically shared by Rhynchonelliformea and Craniiformea (characters 1a, 3, 6, 10d, 12, 13, 16, 17) and 8 characters are shared by Craniiformea and Linguliformea (characters 1b, 5, 8, 9, 10b, 11, 15, 20).
ANSWER: Accordingly to Table 1, at least nine characteristics (1a,3,6,7b,10,11,14,15,17) are similar in craniiforms and rhynchonelliforms, whereas only six features (1b,5,8,9,13,18) are shared by craniiforms and linguliforms.
****************************
COMMENTS IN THE TEXT:
- I know that this is the proper term, but the journal addresses a wider, lesser specialized auditory. I therefore recommend writing with a dorsal and a ventral shell plate, or with two shell plates, or...--------------Correction is DONE
- This is misleading, because in the cited analysis the Brachiopoda also include the phoronids and, thus, do not consist of articulate and inarticulate brachipods only---------------- I have changed the Introduction and added the information about Brachiozoa.
- This is really to short. The author merely cites an abstract here, based on a re-evuation of a published dataset and some data on biomineralization. No developmental data are found herein.----------------------Information is added
- This is nice, but I can not see waht within-group evolutionary trends have in common with the search for sister groups. .----------------------Information is added
- With respect to the previous paragraph it would have been nice to read here what the author expects to find under which premise, like: If Craniidormia were the sister group to Rhynchonelliformis we expect support if ... and so on. Instead this is just a concluding sentence telling, it would be fine to have data. This is not sufficient for a well written introduction.----------------------Information is added
- which buffer did the author use?--------------------- This chapter is supplemented with detailed descriptions
- What are the differences? Thres ist no further hint in the figure subheading. At least one sentence should be added, if the differences are of relevance. .----------------------Information is added
- What does this mean? Is the term ganglion so weakly defined that is can be represented by a nerve and by the way how can a ganglion be represented by something? Does this mean that the nerve is homologous to the subenteric ganglion in other brachiopods? Finally, a supraenteric ganglion is missing, but there are nrueite bundles instead. Why don' these represent the supraenteric ganglion? It is because they are not a "thick nerve", Is a ganglion defined by thickness, and if so, what is the decisive thickness to discriminate a ganglion represented by a thick nerve from a missing ganglion. The author must be much clearer in her description.--------------- I have corrected this phrase. In any case, subenteric ganglion does not have morphology of typical ganglion (see Kuzmina, Temereva, 2021).
- Please add metrical data, like done in the following paragraph-----------DONE
- connective tissue should be replaced by extracellular matrix---------DONE
- This is hardly a methylen-blue staining. Please check thoroughly---------------------- All color photographs have been corrected with Photoshop tool “Auto Color”. After the use of this tool, the color has been changed
- The subheading is not informative and needs an explanation. What does the author want to show. Where is the ancestral state reconstruction based on? What is the blue meandering line? What doe the colours indicate.-------------- Color code and References are added
- I do not understand this. The nerve that connects two groups of perikarya is transformed into the brachial nerve? Where are the two groups of ganglia? This must be explained in more detail.---------------- I do not know where these groups. I have just supposed that juvenile supraenteric ganglion transforms. But this transformation is not traced yet.
- This is a very important piece of information. In which way is the apical organ important, what for, what is its function, is there experimental evidence? If yo, some references must be cited, otherwise it is only concluded from observation. This would be an indirect conclusion, like: It is there and therefore must have an important function. This observation based conclusion, however, does not imply that the function the organ is identical nor that it is important for planktonic life. ------------------- The reference about role of the apical organi in growing larvae is added
- Yes, this either a plesiomorphic or an apomorphic missing of the supraenteric ganglion in juveniles. An outgroup comparison would be helpfull to use the missing supraenteric ganglion in settled juveniles as support for a sistergroup relationship Rhynchonelliformea + Craniiformea. If the outgroups have such a ganglion, than the missing could not support the sister group relationship in question.---------------This comment and similar comments, which suggested to use the outgroup, can not be applied, because it is impossible to chose the correct (!!!!) outgroup.
- exactly this is not shown (see my detailed comment) --------------------- I have added the information into the Conclusion.

Round 2
Reviewer 1 Report
In the revised version of the manuscript “First modern data on the lophophore nervous system in adult Novocrania anomala and a current assessment of brachiopod phylogeny” the author made serious effort and addressed most issues in a satisfactory manner.
However, I am somewhat disappointed about the rather superficial implementation of some of my comments (especially the desired information concerning my major points 1 and 3; e.g. no information is added about the specific differences of juvenile and adult nervous systems, it is just stated that there are differences). But as this additional information would only help to reach a broader audience and make the story a much stronger and coherent one and as this is not crucial for the integrity of the paper, I am generally fine with the changes made.
Finally, I spotted some issues, which are listed below. After consideration of these issues (especially the first point), I recommend this contribution for publication.
Main point – referring to my initial comment and the authors answer:
- Line 312: “The main nerve elements are the ganglia”. That is wrong, a ganglion cannot be part of a nerve (see also my comment concerning terminology issues above).
Authors response: I did not understand this. I did not stay that ganglion is a part of nerve. Nerve elements and nerves are two big difference.
New comment: No, “nerve elements” and “nerves” are more or less the same. A nerve element is part of a nerve and thus axonal origin, and therefore cannot be a ganglion. But maybe it is just a linguistic issue: Do you mean neuronal element? That would work. In its current way it is as wrong as it can be.
Other issues:
Line 55: “…in the frame of for hypothesis…” should probably read “…in the frame of four hypotheses…”?
Line 56: “paraphely” should read “paraphyly”.
Lines 55ff: The changes in the introduction leaves the reader with some unexplained specific terms/clades, e.g. Claciata. Explanation upon first mention would be beneficial.
Lines 89f.: “hypothesis” should read “hypotheses”.
Line 130: Phalloidin has been diluted in which solution?
Lines 404ff: acetylated-alpha-Tubulin labelling: Well, thanks for the explanations in your response letter, but it would be necessary to include your answer in the manuscript, not only in the point-by-point response (other readers will ask the same questions as I did…).
General comment: There are some inconsistencies concerning the taxa names: If written in Latin/Greek, they should be in upper case, if anglicised it is lower case. In the latter, the author uses two wordings, e.g., linguliformes or linguliforms, please unify correctly.
Author Response
Dear Reviewer,
Thank you very much for time that you have devoted to the manuscript and for your useful suggestions and comments. I have addressed all your comments. Please find below my point-by-point answer.
Beast regards,
Elena Temereva
RESPONSE TO REVIEWER 1_Round-2:
However, I am somewhat disappointed about the rather superficial implementation of some of my comments (especially the desired information concerning my major points 1 and 3; e.g. no information is added about the specific differences of juvenile and adult nervous systems, it is just stated that there are differences). But as this additional information would only help to reach a broader audience and make the story a much stronger and coherent one and as this is not crucial for the integrity of the paper, I am generally fine with the changes made.------------ I have added the information about the difference between adult and juvenile nervous system.
Your point 1 was about lophophorates. The lophophorates is not central group of this study. I have mentioned about lophophorates, because there are similarities in ideas about evolution of the lophophore in brachiopods and in other lophophorates – phoronids and bryozoans. I am not planning to include all lophophorates in this study.
Line 312: “The main nerve elements are the ganglia”. That is wrong, a ganglion cannot be part of a nerve (see also my comment concerning terminology issues above). Authors response: I did not understand this. I did not stay that ganglion is a part of nerve. Nerve elements and nerves are two big difference. New comment: No, “nerve elements” and “nerves” are more or less the same. A nerve element is part of a nerve and thus axonal origin, and therefore cannot be a ganglion. But maybe it is just a linguistic issue: Do you mean neuronal element? That would work. In its current way it is as wrong as it can be.------------------ Thank you very much. Yes, you are right. It should be neuronal elements. I have changed the text.
Line 55: “…in the frame of for hypothesis…” should probably read “…in the frame of four hypotheses…”? ---------------- Yes. Correction is done
Line 56: “paraphely” should read “paraphyly”.-------------Corrected
Lines 55ff: The changes in the introduction leaves the reader with some unexplained specific terms/clades, e.g. Claciata. Explanation upon first mention would be beneficial---------------- I have added the explanation.
Lines 89f.: “hypothesis” should read “hypotheses”-----------Corrected
Line 130: Phalloidin has been diluted in which solution? ------------------in PBT. This added into the text
Lines 404ff: acetylated-alpha-Tubulin labelling: Well, thanks for the explanations in your response letter, but it would be necessary to include your answer in the manuscript, not only in the point-by-point response (other readers will ask the same questions as I did…).---------Because I do not have answer, I added my idea about this specificity.
General comment: There are some inconsistencies concerning the taxa names: If written in Latin/Greek, they should be in upper case, if anglicised it is lower case. In the latter, the author uses two wordings, e.g., linguliformes or linguliforms, please unify correctly.-------Yes. You are completely right. I have done all corrections.
Reviewer 2 Report
Dear Author,
thanks for the credits. I still have a problem with the number of characters in support for a sister group relationship between craniiforms and rhynchonelliforms. In terms of cladistics an outgroup is missing, whereby I know that this is a problem. Even on does not use an outgroup, characters 10, 11,14,15 are not logically independent of character 3. If a second accessory nerve is not present, this nerve cannot be connected to other nerves, simply because it is absent. Thu, not nine, but only five characters support the sister group relationship in question. The logical dependence of characters 10, 11, 14,15 and character 3 is obvious to everybody who is familiar with character coding. You presently just sum up characters without any evaluation - which is presently hardly possible due to the unknown relationships. I am writing this just to call for being a little bit more cautious in the conclusions; the support is not as evident as it seems just by counting the number of identical structures.
Author Response
ANSWER: Dear Reviewer, thank you very much for your comments. You are right; these characteristics must be analyzed carefully. However, at this moment, we do not have enough information to make comprehensive analysis. We need more information from Linguliformea and Craniiformea, because only one species for each group is studied by modern methods. At this moment, we can see that the organization of the CNS (without innervation of tentacles) is similar between Craniiformea and Rhynchonelliformea. I have added this into the text.